# Reachability Weighted Offline Goal-conditioned Resampling

## Abstract

Offline goal-conditioned reinforcement learning (RL) relies on fixed datasets where many potential goals share the same state and action spaces. However, these potential goals are not explicitly represented in the collected trajectories. To learn a generalizable goal-conditioned policy, it is common to sample goals and state–action pairs uniformly using dynamic programming methods such as Q-learning. Uniform sampling, however, requires an intractably large dataset to cover all possible combinations and creates many unreachable state–goal–action pairs that degrade policy performance. Our key insight is that sampling should favor transitions that enable goal achievement. To this end, we propose Reachability Weighted Sampling (RWS). RWS uses a reachability classifier trained via positive–unlabeled (PU) learning on goal-conditioned state–action values. The classifier maps these values to a reachability score, which is then used as a sampling priority. RWS is a plug-and-play module that integrates seamlessly with standard offline RL algorithms. Experiments on six complex simulated robotic manipulation tasks, including those with a robot arm and a dexterous hand, show that RWS significantly improves performance. In one notable case, performance on the HandBlock-Z task improved by nearly 50% relative to the baseline. These results indicate the effectiveness of reachability-weighted sampling.

## 1 Introduction

Goal-conditioned reinforcement learning (GCRL) has emerged as a promising framework for training agents to master diverse skills in complex environments (Laird et al., 1987; Kaelbling, 1993; Liu et al., 2022). Unlike conventional reinforcement learning methods that rely on carefully shaped rewards, GCRL typically uses sparse rewards that merely indicate whether a goal has been reached, thereby simplifying the feedback mechanism without compromising the agent's ability to optimize its behavior. Recent efforts have also connected GCRL with unsupervised contrastive learning (Eysenbach et al., 2022; Zheng et al., 2023), linking goal-driven approaches to domains like computer vision and natural language processing. Moreover, the flexibility of goal representations ranging from state coordinates to images or linguistic descriptions broadens the applicability of GCRL. By learning these goals, agents can acquire meta-skills conducive to hierarchical control and planning for more complex tasks (Ghosh et al., 2023; Wang et al., 2023a; Li et al., 2022). Offline GCRL extends this paradigm by combining goal-conditioned tasks with offline reinforcement learning, which eliminates the need for additional environment interactions and leverages static datasets for policy learning (Levine et al., 2020; Agarwal et al., 2023; Liu et al., 2021). This is particularly advantageous in scenarios where active exploration is expensive or unsafe. As datasets containing diverse goals become increasingly available (Eysenbach et al., 2022; Zheng et al., 2023), policies trained in offline GCRL can gain a wide range of reusable primitives for downstream tasks (Ghosh et al., 2023; Wang et al., 2023a; Li et al., 2022). Nevertheless, effectively learning multi-goal behaviors from purely offline data remains challenging due to the limited coverage of state-goal pairs and the prevalence of suboptimal or noisy samples. It is common to randomly combine state-action pair with random goals to learn a generalizable goal-conditioned policy from a static dataset (Chebotar et al., 2021; Eysenbach et al., 2022), however, random goal sampling tends to decrease the policy performance as it generates suboptimal even non-optimal data pairs Park et al. (2024); Eysenbach et al. (2020).

In this work, we propose a novel sampling strategy, Reachability-Weighted Sampling (RWS), to address challenges in offline GCRL. As illustrated in Figure 3, RWS employs a reachability classifier that evaluates

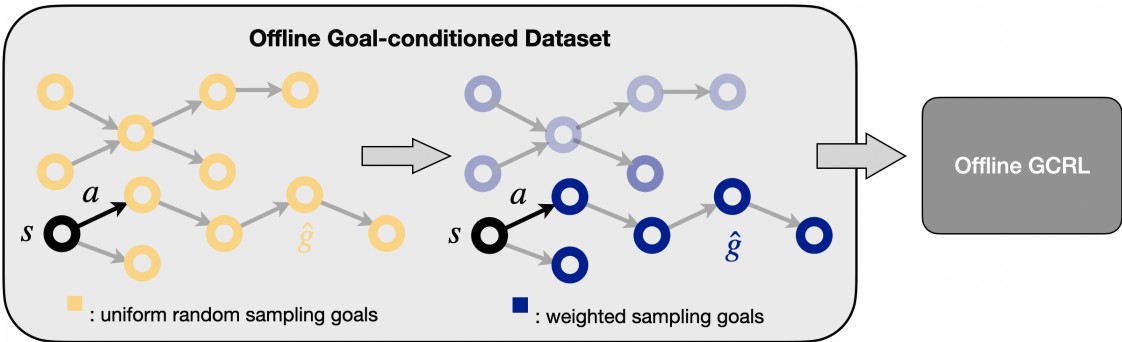

Figure 1: **Diagram of Reachability Weighted Sampling (RWS)**. The key idea of RWS is to learn a reachability classification function that samples potentially reachable goals (the dark blue circles in the figure) more frequently instead of sampling all goals in the dataset uniformly (the light yellow circles).

whether a goal is achievable from a state-action pair by leveraging the goal-conditioned Q-value. We form a positive-unlabeled dataset by generating positive examples via hindsight relabeling and unlabeled examples via uniform goal sampling from the offline dataset. The classifier is optimized with non-negative PU learning to convert Q-values into reachability scores. An exponential transformation and subsequent normalization yield sampling weights over the goals. Consequently, transitions with potentially reachable goals are sampled more frequently, while unreachable ones are down-weighted. This resampling strategy enhances the quality of goal-conditioned experiences, reducing the impact of noisy, suboptimal samples.

The main contribution of this work can be summarized as: 1) we present a novel, plug-and-play resampling scheme for offline goal-conditioned reinforcement learning. Our method leverages a Q-value-based reachability classifier, trained using PU learning, to re-weight experience by preferentially sampling transitions with achievable goals. 2) We integrate RWS with standard offline RL algorithms (e.g., MCQ, TD3BC, ReBRAC) and demonstrate that our approach consistently enhances both performance and data efficiency.

## 2 Related Works

**Goal-Conditioned Augmentations** A classical technique in goal-conditioned reinforcement learning is Hindsight Experience Replay (HER) (Andrychowicz et al., 2017), which replaces task goals with those corresponding to states actually visited during execution. HER enriches the reward signal, facilitating value function learning and enabling the extraction of useful skills even from trajectories that would otherwise yield zero rewards. Building on HER, goal-conditioned weighted supervised learning methods (Liu et al., 2021; Yang et al., 2022a; Ma et al., 2022; Hejna et al., 2023b; Lei et al., 2024) repeatedly perform weighted imitation learning with relabeled trajectories. However, HER tends to consistently generate successful trajectories, which leads to a bias in the training samples. In contrast, the Actionable Model (AM) (Chebotar et al., 2021) randomly selects goals from the entire dataset to augment goal-conditioned trajectories. This offline goal-conditioned relabeling not only significantly increases the dataset size but also produces many unsuccessful trajectories, as the relabeled goal $g'$ is often sampled from states not encountered in the current trajectory. To sample reachable goals, recent works (Huang et al., 2024; Mezghani et al., 2023) construct a graph from the offline data and use graph search to identify potentially reachable goals, thereby generating more effective goal-conditioned trajectories or augmenting the reward to train a more generalizable policy. However, building a directed graph becomes impractical when the state space is high-dimensional or the dataset is large. Moreover, random goal sampling may drastically decrease data quality, making it difficult to learn value functions for policy optimization (Park et al., 2024; Eysenbach et al., 2020).

**Weighted Experience Resampling** The effect of dataset quality on offline reinforcement learning (RL) policy learning has been studied in previous works (Hong et al., 2023; Park et al., 2024; Zhan et al., 2022). A key observation is that policies may mimic suboptimal actions, which degrades performance. One solution is to modify the sampling strategy so that the policy is constrained to high-quality state-action pairs rather

than all pairs in the dataset (Hong et al., 2023). An intuitive approach is advantage weighting, which uses an advantage function to identify state-action pairs that are likely to yield better performance (Peng et al., 2019; Wang et al., 2020). Another well-known technique is density-ratio importance correction estimation (DiCE) (Nachum et al., 2019a), primarily used for policy evaluation. Recent works (Lee et al., 2021; Nachum et al., 2019b; Rashidinejad et al., 2022; Zhan et al., 2022) apply DiCE to re-weight behavior cloning, estimating the sampling weights of offline data. The Density-Weighting (DW) method proposed by Hong et al. (2023) decomposes the DiCE training from policy optimization, making it more flexible for integration with any offline RL method. In goal-conditioned settings, recent studies (Agarwal et al., 2023; Jain & Unhelkar, 2024) have proposed goal-conditioned DiCE variants to assist policy training with goal-conditioned rewards. However, these approaches either target general offline RL methods without addressing the random goal sampling issue or fail to consider the goal stitching ability (Agarwal et al., 2023). In this work, we propose a weighted goal sampling strategy that focuses on sampling potentially reachable goals rather than all possible state-goal-action pairs in the dataset.

## 3 Background

### 3.1 Goal-Conditioned Reinforcement Learning

Goal-Conditioned Reinforcement Learning (GCRL) (Liu et al., 2022) enables agents to learn policies for achieving diverse goals. Formally, a Goal-Conditioned Markov Decision Process (GCMDP) is defined as $\mathcal{M} = (\mathcal{S}, \mathcal{A}, \mathcal{G}, P, r, \gamma)$, where $\mathcal{S}$ is the state space, $\mathcal{A}$ the action space, $\mathcal{G}$ the goal space, $P(s_{t+1}|s_t, a_t)$ the transition dynamics, $r$ the reward function, and $\gamma \in [0, 1)$ the discount factor. The sparse reward function employed in GCRL can be expressed as:

$$r(s_t, a_t, g) = \begin{cases} 0, & \|\phi(s_t) - g\|_2^2 < \delta \\ -1, & \text{otherwise} \end{cases} \tag{1}$$

where $\delta$ is a threshold and $\phi : \mathcal{S} \to \mathcal{G}$ is a known state-to-goal mapping function. The agent learns a goal-conditioned policy $\pi(a|s, g) : \mathcal{S} \times \mathcal{G} \to \Delta(\mathcal{A})$, mapping state-goal pairs to action distributions. The objective is to maximize:

$$\mathcal{J}(\pi) = \mathbb{E}_{g \sim P_g, s_0 \sim P_0, a_t \sim \pi(\cdot|s_t, g), s_{t+1} \sim P(\cdot|s_t, a_t)} \left[ \sum_{t=0}^{T} \gamma^t r(s_t, a_t, g) \right], \tag{2}$$

where $P_g$ is the goal distribution, $P_0$ the initial state distribution, and $T$ the horizon. This formulation enables agents to learn generalizable behaviors that can be composed to solve complex tasks through appropriate goal specification. The goal-conditioned value function $Q^\pi(s, g, a) = \mathbb{E}_\pi[\sum_{t=0}^{T} \gamma^t r(s_t, a_t, g)|s_t = s, a_t = a]$ can be described in two ways: 1) as a distance-measuring function, which indicates the expected number of policy steps required to reach goals (Wang et al., 2023b; Hejna et al., 2023a), or 2) as a scaled probability mass function, estimating the future state density of a goal-conditioned policy (Eysenbach et al., 2020; 2022; Zheng et al., 2023). When viewed as a distance-measuring function, trajectories terminate upon goal achievement by the policy, whereas the probability mass function perspective does not make this assumption.

### 3.2 Offline Reinforcement Learning

In the offline GCRL setting (Park et al., 2024), the agent learns only from an offline dataset $\mathcal{D}$ without interacting with the environment. The dataset $\mathcal{D}$ consists of $N$ goal-conditioned trajectories:

$$\mathcal{D} := \{\tau_i = (s_0^i, a_0^i, r_0^i, \ldots \mid g^i)\}_{i=1}^N.$$

The dataset $\mathcal{D}$ is typically generated by unknown policies. Commonly, offline RL methods focus on estimating the return of a policy $\pi$ using techniques such as Q-learning or actor-critic algorithms, relying solely on batches of state-action pairs $(s_t, a_t)$ sampled from $\mathcal{D}$. Consequently, the estimated return $\mathcal{J}_\mathcal{D}(\pi)$ derived solely from dataset $\mathcal{D}$ often suffers from inaccuracies, especially when the policy $\pi$ encounters states and actions not adequately represented in the dataset $\mathcal{D}$. To mitigate this issue, offline RL algorithms typically employ pessimistic or conservative regularization techniques Levine et al. (2020); Yao et al.; Hong et al.

(2023). These methods penalize deviations of the learned policy $\pi$ from the behavior policy $\pi_{\mathcal{D}}$, restricting $\pi$ from generating actions that are not present in $\mathcal{D}$. In goal-conditioned settings, the optimization objective for these algorithms can be expressed as:

$$\max_{\pi} \mathcal{J}_{\mathcal{D}}(\pi) - \alpha \mathbb{E}_{(s_t, g, a_t) \sim \mathcal{D}} \left[ \mathcal{C}(\pi(s_t, g), a_t) \right], \tag{3}$$

where $\mathcal{C}(\cdot, \cdot)$ denotes a divergence measure (e.g., Kullback–Leibler divergence (Kullback, 1951)) between the learned policy $\pi$ and the behavior policy $\pi_{\mathcal{D}}$, and $\alpha \in \mathbb{R}^+$ controls the regularization strength. This approach ensures robust policy learning despite limitations inherent in offline datasets.

### 3.3 Positive-Unlabeled Learning.

For a binary classification problem with input instances $x \in \mathbb{R}^n$ and binary labels $y \in \{0, 1\}$. Typically, labeled datasets are available for both positive and negative classes, denoted respectively as $D_P$ (positive dataset) and $D_N$ (negative dataset). Additionally, let $\eta_p = \Pr(y = 1)$ represent the class prior probability for the positive class, and $\eta_n = \Pr(y = 0) = 1 - \eta_p$ for the negative class. The standard cross-entropy loss commonly used for binary classification can then be expressed as:

$$\mathcal{L} = \eta_p \mathbb{E}_{(x,y) \sim D_P} [- \log \hat{y}] + \eta_n \mathbb{E}_{(x,y) \sim D_N} [- \log(1 - \hat{y})],$$

where $\hat{y} = c(x)$ is the discriminator's predicted probability of the positive class. In the Positive-Unlabeled (PU) learning scenario (Elkan & Noto, 2008), only a positively labeled dataset $D_P$ and an unlabeled dataset $D_U$ are available. The unlabeled dataset $D_U$ is a mixture containing unknown proportions of positive and negative instances. A naive approach might consider all unlabeled instances as negative, but this introduces systematic bias, incorrectly penalizing positive instances within $D_U$. To correct this bias, existing methods leverage the assumption that the unlabeled dataset $D_U$ is a mixture of positive and negative samples with proportions $\eta_p$ and $\eta_n$, respectively. This allows the expectation for unlabeled data to be decomposed as (du Plessis et al., 2014):

$$\mathbb{E}_{(x,y) \sim D_U} [- \log(1 - \hat{y})] = \eta_p \mathbb{E}_{(x,y) \sim D_P} [- \log(1 - \hat{y})] + \eta_n \mathbb{E}_{(x,y) \sim D_N} [- \log(1 - \hat{y})].$$

Applying this decomposition, the original cross-entropy loss can be reformulated as:

$$\mathcal{L}_{PU} = - \left[ \eta_p \mathbb{E}_{(x,y) \sim D_P} [\log \hat{y}] + \mathbb{E}_{(x,y) \sim D_U} [\log(1 - \hat{y})] - \eta_p \mathbb{E}_{(x,y) \sim D_P} [\log(1 - \hat{y})] \right].$$

However, (Kiryo et al., 2017) emphasize a critical refinement: the term $\mathbb{E}_{(x,y) \sim D_U} [\log(1 - \hat{y})] - \eta_p \mathbb{E}_{(x,y) \sim D_P} [\log(1 - \hat{y})]$ must remain non-negative to avoid severe overfitting and instability during training. Thus, introducing a max operator ensures this condition is explicitly enforced, yielding the non-negative PU learning:

$$\mathcal{L}_{PU} = - \left[ \eta_p \mathbb{E}_{(x,y) \sim D_P} [\log \hat{y}] + \max \left( \mathbb{E}_{(x,y) \sim D_U} [\log(1 - \hat{y})] - \eta_p \mathbb{E}_{(x,y) \sim D_P} [\log(1 - \hat{y})], \ 0 \right) \right].$$

### 3.4 Motivation: Challenges of Offline GCRL

In this section, we address two primary challenges encountered in offline Goal-Conditioned Reinforcement Learning (GCRL), which motivate the methodological design choices proposed in this paper.

**Goal Stitching.** A significant challenge in offline goal-conditioned RL is stitching together the initial and final states of different trajectories to learn more diverse behaviors. In offline GCRL, the agent learns from a static dataset, where each trajectory is conditioned on a specific goal. Consequently, datasets often lack complete demonstrations from a starting state to all possible goals. This scarcity makes it difficult for an agent to directly learn a policy capable of reaching every goal. An ideal goal-conditioned policy should exhibit *goal stitching*, the ability to combine segments from different trajectories to form novel paths not explicitly demonstrated in the original dataset. Dynamic programming approaches, such as Q-learning, naturally support goal stitching through their value iteration process by randomly selecting goals from the

dataset and relabeling them using existing trajectories. For example, in Fig. 2, swapping the goals $g^\lambda$ and $g^\eta$ between trajectories $\tau^\lambda$ and $\tau^\eta$ enables the agent to reach $g^\lambda$ from $s_0^\eta$ and $g^\eta$ from $s_0^\lambda$. A general goal relabeling strategy can be described as:

$$(s_i^\tau, a_i^\tau, \hat{g}) \quad \text{with} \quad (s^\tau, a^\tau) \sim \tau, \hat{g} \sim P_{\mathcal{D}(g)}, \tag{4}$$

where $(s^\tau, a^\tau) \sim \tau$ denotes the state-action pair from trajectory $\tau$, and $\hat{g} \sim P_{\mathcal{D}(g)}$ denotes uniformly sampling a goal $\hat{g}$ from the offline dataset $\mathcal{D}$.

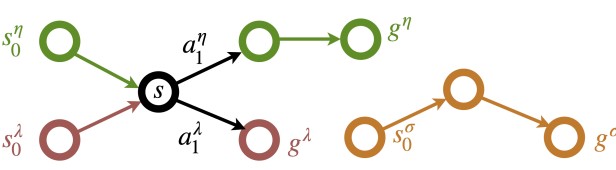

Figure 2: A simple illustration of an offline GCRL dataset with three goal-conditioned trajectories: $\mathcal{D} = \{\tau_{s_0^\eta \to g^\eta}^\eta, \tau_{s_0^\lambda \to g^\lambda}^\lambda, \tau_{s_0^\sigma \to g^\sigma}^\sigma\}$.

However, uniformly random goal sampling and relabeling can potentially create unreachable goal-state pairs. As illustrated in Fig. 2, the state $s^\sigma$ can only reach states along trajectory $\tau^\sigma$, whereas uniformly random goal sampling predominantly generates unreachable state-goal pairs for learning. Since the random goal sampling strategy tends to produce negative training pairs, it may hinder the training of the goal-conditioned policy and even result in worse performance compared to approaches that do not utilize random goal sampling (Park et al., 2024).

**Unnecessary Conservativeness of Suboptimal Data** In offline GCRL, combining the random goal sampling strategy with Eq. 3 results in poor policy performance (Park et al., 2024). This occurs because random goal sampling generates noisy and suboptimal goal-state-action pairs. For instance, in Fig. 2, augmenting the original goal-state-action pair $(s, a_1^\lambda, g^\lambda)$ from trajectory $\tau^\lambda$ with the goal $g^\eta$ creates a suboptimal pair $(s, a_1^\lambda, g^\eta)$, even though both $g^\eta$ and $g^\lambda$ are reachable from state $s$. Due to the policy imitation constraint term $\mathcal{C}(\cdot, \cdot)$ in Eq. 3, random goal sampling induces low-performance trajectories, leading the policy to imitate actions from these inferior trajectories. Prior empirical studies indicate that when datasets predominantly contain suboptimal trajectories, state-of-the-art offline RL algorithms often fail to substantially surpass the average performance of the dataset trajectories (Hong et al., 2023; Yao et al.).

In summary, relabeling state-action transitions with random goals is necessary to learn generalizable goal-conditioned policies. However, uniformly random goal sampling frequently generates low-quality goal-state-action pairs, which negatively impact policy performance (Park et al., 2024; Hong et al., 2023; Yao et al.). Ideally, a goal relabeling strategy should preferentially sample potentially reachable goals rather than uniformly sampling all goals in the dataset.

## 4 Methods: Reachability-Weighted Sampling

In this work, we propose a weighted goal-sampling strategy for offline GCRL training. Our strategy samples reachable goals more frequently than uniformly random goals. Inspired by positive-unlabeled (PU) learning, we train a reachability classifier using goal-conditioned state-action values as discriminative features. Based on the trained classifier, we define a sampling weight to re-weight uniformly sampled goals for offline GCRL policy training, allowing potentially reachable goal-state-action pairs to be sampled more frequently.

### 4.1 Reachability Identification by PU Learning

The key idea of our method is to filter positive-reachable state-goal-action pairs $(s, g, a)$ from unlabeled data (described in Eq. 4) by training a reachability classifier. The positive-unlabeled reachability dataset consists of two parts:

- **Positive data:** For any two states $s_i, s_h$ in a trajectory $\tau$ with $i < h$, we know the goal $g_h = \phi(s_h)$ can be reached from $s_i$ in $(h - i)$ steps. Using the hindsight relabeling technique (Andrychowicz et al., 2017), we generate a relabeled dataset: $\mathcal{D}_P = \{(s_i, a_i, g_h)\}_{n=1}^{n^p}$, containing known reachable state-goal-action pairs.

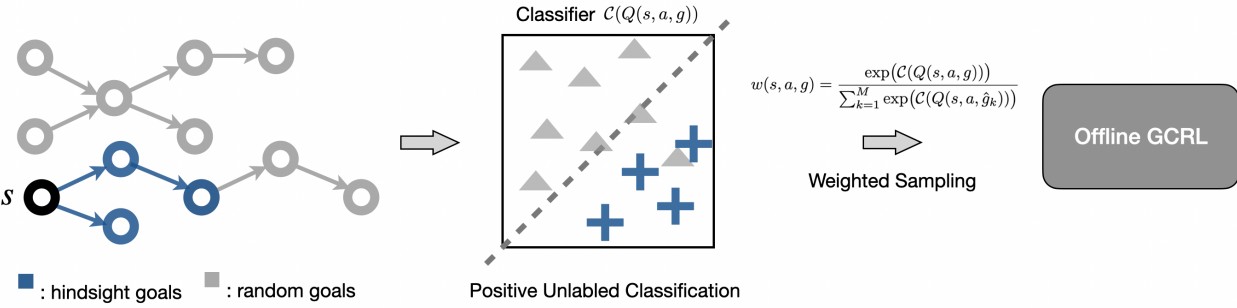

Figure 3: **Reachability Weighted Sampling**: An overview of our framework. We create a positive-unlabeled dataset by applying hindsight relabeling to produce positive data and random goal sampling to produce unlabeled data. A PU learning-based binary classifier $\mathcal{C}$ (using the goal-conditioned Q-value as the discriminative feature) is then trained to identify the reachable goals. Finally, we apply $\mathcal{C}$ to a weighting function so that offline GCRL can sample potentially reachable goals and refine goal-conditioned policy learning.

- **Unlabeled data:** We consider the dataset constructed by uniform random goal sampling (Eq. 4) as unlabeled data. For any two states $s_i, s_j$ in dataset $\mathcal{D}$, it is unknown whether goal $g_j = \phi(s_j)$ can be reached from $s_i$. Therefore, pairs $(s_i, a_i, g_j)$ form unlabeled data, representing a mixture of reachable and non-reachable examples: $\mathcal{D}_U = \{(s_i, a_i, g_j)\}_{m=1}^{m^U}$.

In our goal-conditioned task setting, the reward is defined as in Eq. 1, and task trajectories terminate once the policy achieves the goal. Since the sparse reward function depends solely on whether $\phi(s) = g$, a policy's return is determined by the number of time steps spent reaching the goal. Consequently, the goal-conditioned value function $Q^\pi(s, g, a)$ represents the expected number of steps for policy $\pi$ to reach the goal. Specifically, $Q : S \times G \times A \to (-\infty, 0], \quad q = Q(s, g, a), \quad q \leq 0$. Thus, higher (i.e., less negative) values of $Q^\pi(s, g, a)$ indicate that fewer steps are needed to achieve the goal. This property allows the state-goal-action value $q$ to naturally capture reachability discrepancies, making it an effective signal for classification.

We formulate a binary classification problem with input space $S \times G \times A$. Labels are $Y = +1$ for positive (reachable) examples and $Y = -1$ for negative (non-reachable) examples. Using positive data $\mathcal{D}_P$ and unlabeled data $\mathcal{D}_U$, we train a classifier: $f : S \times G \times A \to \{+1, -1\}$. Furthermore, classifier $f$ is defined as the composition of the goal-conditioned value function $Q$ and a binary classifier, which is given by: $f(s, g, a) = C(Q(s, g, a))$. Given a pretrained goal-conditioned value function $Q_\psi(s, g, a)$, we directly apply logistic regression on $q = Q_\psi(s, g, a)$ to learn the reachability classifier. It is optimized using a non-negative PU learning loss:

$$\mathcal{L}_{PU}(\theta) = -\left[ \eta_p \, \mathbb{E}_{(s,g,a) \sim \mathcal{D}_P}[\log C_\theta(Q_\psi(s, g, a))] + \max\left\{ \mathbb{E}_{(\hat{s},\hat{g},\hat{a}) \sim \mathcal{D}_U}[\log(1 - C_\theta(Q_\psi(\hat{s}, \hat{g}, \hat{a})))] \right. \right.$$
$$\left. \left. -\eta_p \, \mathbb{E}_{(s,g,a) \sim \mathcal{D}_P}[\log(1 - C_\theta(Q_\psi(s, g, a)))], 0 \right\} \right]. \tag{5}$$

Here, $C_\theta(x)$ is a linear logistic classifier parameterized by $\theta$, and $\eta_p$ is a weighting coefficient interpreted as the prior probability of the positive class (practically, we set $\eta_p = 0.5$). This coefficient balances the contribution of positive examples in risk estimation. In this work, we do not aim to accurately identify reachability but instead seek a classifier that optimistically predicts higher scores for potentially reachable goals. Preferring false positives is acceptable to avoid ignoring potentially reachable state-action-goal pairs.

## 4.2   Reachability Weighted Sampling

Our motivation is to sample all potentially reachable state-goal-action pairs equally, regardless of the distance between the state $s$ and the goal $g$. For example, goals expected to be reached in 2 steps and goals expected

to be reached in 10 steps should be sampled equally. At the same time, unreachable goals should be sampled as little as possible. With a trained classifier $C_\theta$ (a simple linear logistic classifier), the classification score becomes proportional to the goal-conditioned Q-value, satisfying the stated requirement.

Figure 4 illustrates the learned reachability classifier. The dark blue dots on the x-axis represent Q-values of positive state-goal-action pairs sampled from $\mathcal{D}_P$. The red dots on the x-axis represent Q-values of negative pairs sampled from $\mathcal{D}_U$. The dark green curve sigmoid curve indicate the outcomes of PU learning—the binary classification probabilities. Reachable state-goal-action pairs are assigned classification scores close to 1, while scores assigned to state-goal-action pairs decrease toward 0 as reachability decreases. One can simply associate the classification score $c = C_\theta(Q_\psi(s, \hat{g}, a))$ to $(s, \hat{g}, a)$ where the goal $\hat{g}$ is uniformly randomly sampled from the dataset $\mathcal{D}$:

$$w(s, \hat{g}, a) = \frac{\mathcal{C}(Q(s, \hat{g}, a))}{\int_{g' \in \mathcal{D}} \mathcal{C}(Q(s, a, g'))}.$$

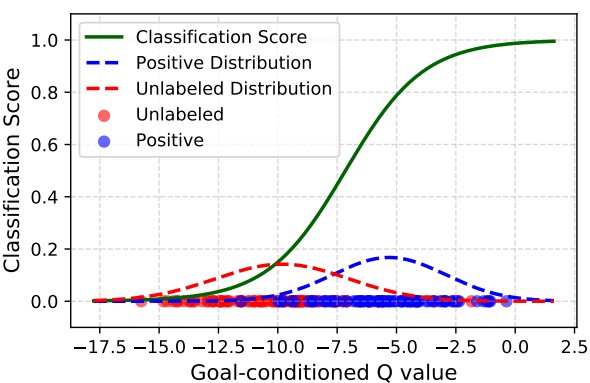

Figure 4: **Reachability Classifier Based on Goal-Conditioned Q value**: The plot illustrates the relationship between the goal-conditioned Q-value and the classification score. The binary classification probabilities learned through PU learning, effectively mapping Q-values to reachability scores.

However, due to the approximation error of the pretrained $Q_\psi(s, g, a)$, it may estimate a low value for some reachable $(s, g, a)$ pairs that are potentially reachable Eysenbach et al. (2020). As a consequence, the classifier $C_\theta(Q)$ may assign low or even zero weight to $(s, g, a)$, which limits the ability to train a generalizable goal-conditioned policy. To avoid assigning the sampling weight to zero, we adjust the sampling weight with an exponential function:

$$w(s, g, a) = \frac{\exp \mathcal{C}_\theta(Q_\psi(s, g, a))}{\int_{g' \in \mathcal{D}} \exp \mathcal{C}_\theta(Q_\psi(s, a, g'))} \approx \frac{\exp \mathcal{C}_\theta(Q_\psi(s, g, a))}{\frac{1}{N} \sum_{\substack{k=1 \\ g'_k \sim \mathcal{D}}}^{N} \exp \mathcal{C}_\theta(Q_\psi(s, a, g'_k))}. \tag{6}$$

By using this sampling weight, we can sample potentially reachable goals more frequently, regardless of the expected distance between the state and the goal, while having a lower probability of sampling unreachable $(s, g, a)$ pairs.

### 4.3 Practical Implementation

---

**Algorithm 1** Reachability Weighted Sampling with Generic Offline Goal-Conditioned RL Algorithms

---

1: **Input:** Goal-conditioned dataset $\mathcal{D}$
2: Initialize policy $\pi$, value network $Q_\psi$, and classifier $C_\theta$.
3: **while** not converged **do**
4:    Sample a batch $\mathcal{B}$ of $N$ tuples $(s, a, s')$ from $\mathcal{D}$.
5:    Create a positive batch $\mathcal{B}_P = \{(s, g_h, a)\}_N$ by obtaining hindsight goals $g_h$ from $\mathcal{B}$ using HER.
6:    Create an unlabeled batch $\mathcal{B}_U = \{(s, \hat{g}, a)\}_N$ by randomly sampling $N$ goals $\hat{g}$ from $\mathcal{D}$.
7:    Update $C_\theta$ with batches $\mathcal{B}_P$ and $\mathcal{B}_U$ using Eq. 5, keeping $Q_\psi$ frozen.
8:    Compute next states and calculate the goal-conditioned reward using Eq. 1 for $\mathcal{B}_U$, forming $\mathcal{B}_\pi = \{(s, a, \hat{g}, r, s')\}_N$.
9:    Compute the sampling weight $w(s, \hat{g}, a)$ for $\mathcal{B}_\pi$ using Eq. 6.
10:    Update policy $\pi$ and value network $Q_\psi$ with the weights $w(s, \hat{g}, a)$ and batch $\mathcal{B}_\pi$ using any offline GCRL objective.
11: **end while**

---

As the reachability classifier depends on the goal-conditioned Q value, this suggests a two-stage approach: 1) Train the goal-conditioned value function using offline RL algorithms until convergence. 2) Freeze the value

function network weights and train a reachability classifier using PU learning. However, this approach will introduces an additional hyperparameter (the number of pretraining iterations ) and can be computationally intensive and time-inefficient. To mitigate these issues, we opt to train the reachability classifier $\mathcal{C}(1|s, g, a)$ concurrently with the offline RL algorithm (i.e., value functions and policy). In our practical implementation, we alternate between one iteration of offline RL update and one iteration of of linear logistic classifier $C_\theta$ update. The training procedure is outlined in Algorithm 1.

## 5 Experiments

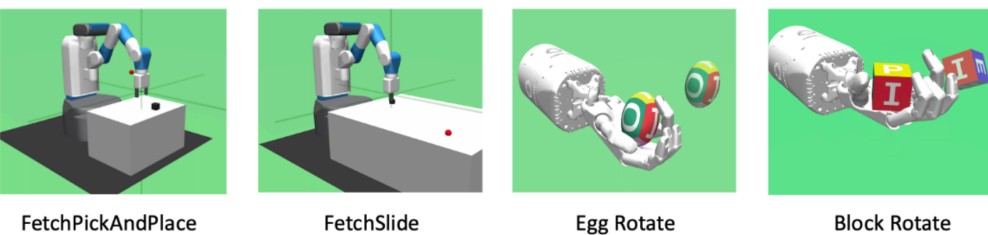

Figure 5: The goal-conditioned tasks selected for experiments in this work.

We begin by introducing the benchmarks and baseline methods, followed by a detailed description of the experimental setup. Next, we present the results and analysis, which evaluate tthe effectiveness of reachability weighted sampling (RWS) for offline GCRL problems.

**Tasks**   Six goal-conditioned tasks from Plappert et al., 2018; Yang et al., 2022b are used in our experiments. These tasks include two fetching manipulation tasks (FetchPickAndPlace, FetchPush) and four dexterous in-hand manipulation tasks (HandBlock-Z, HandBlock-Parallel, HandBlock-XYZ, HandEggRotate). In the fetching tasks, the virtual robot must move an object to a target position within the virtual space. In the in-hand manipulation tasks, the agent is required to rotate the object to a specified pose. The offline dataset for each task consists of a mixture of expert trajectories and random trajectories. The fetching tasks and their datasets are adopted from Plappert et al., 2018. The expert and random trajectories for the in-hand manipulation tasks are generated following the procedure in Yang et al., 2022b.

Our experimental study investigates Reachability Weighted Sampling (RWS) from several perspectives: 1)**Goal-Conditioned Performance:** We begin with a toy bandit experiment to illustrate the impact of prioritized sampling on behavior policies. 2) **Data Efficiency:** We examine how RWS improves data utilization by evaluating datasets with varying expert demonstration ratios. 3) **Comparison to Existing Methods:** We compare the priority function of RWS with those used in existing resampling methods for goal-conditioned offline RL (e.g., GOFAR (Agarwal et al., 2023)) and for non-goal-conditioned settings (e.g., DW, AW (Hong et al., 2023; Peng et al., 2019)). Furthermore, we compare offline RL algorithms that integrate RWS with their standard counterparts to demonstrate improvements in goal-conditioned skill learning.

### 5.1 Goal-Conditioned Task Performances

Table 1 reports the performance of various offline goal-conditioned RL methods, both with and without the integration of Reachability Weighted Sampling (RWS). Each algorithm was trained with 10 random seeds, and higher scores indicate better policy performance.

The table compares GCRL-oriented methods such as GoFAR, WGCSL, and GCSL alongside goal-conditioned variants of offline RL methods (GC-TD3BC, GC-ReBRAC, GC-MCQ, GC-CRR, GC-IQL) with and without RWS. The RWS-enhanced methods consistently deliver improved results. For instance, in the FetchPush task, GC-TD3BC+RWS achieves a mean score of 34.19, compared to 28.16 for GC-TD3BC. Similarly, in HandBlock-Z, GC-TD3BC+RWS scores 49.43, surpassing GC-TD3BC's 32.81. The bold values in the table indicate the best results based on a T-test with a p-value < 0.05. These findings demonstrate that incor-

|  | FetchPush | FetchPick | HandBlock-Z | HandBlock-Parallel | HandBlock-XYZ | HandEggRotate |
|---|---|---|---|---|---|---|
| GC-TD3BC | 28.16 ± 3.17 | 28.18 ± 3.40 | 32.81 ± 7.87 | 15.24 ± 5.27 | 14.64 ± 4.64 | 34.00 ± 6.23 |
| GC-ReBRAC | 22.77 ± 3.54 | 25.05 ± 3.32 | 28.78 ± 7.30 | 12.50 ± 4.58 | 6.46 ± 2.78 | 19.95 ± 5.21 |
| GC-MCQ | **37.95 ± 1.89** | 31.42 ± 3.03 | 18.80 ± 7.30 | 5.09 ± 3.61 | 0.69 ± 1.28 | 11.44 ± 4.51 |
| GC-CRR | 21.22 ± 3.20 | 15.56 ± 3.10 | 0.64 ± 0.65 | 0.09 ± 0.25 | 0.03 ± 0.10 | 0.02 ± 0.08 |
| GC-IQL | 3.03 ± 2.49 | 1.41 ± 1.63 | 0.90 ± 1.13 | 0.03 ± 0.14 | 0.07 ± 0.29 | 0.02 ± 0.08 |
| GoFAR | 21.32 ± 5.21 | 18.64 ± 3.32 | 5.26 ± 8.32 | 0.01 ± 0.09 | 0.95 ± 5.01 | 4.7 ± 6.83 |
| WGCSL | 10.51 ± 6.62 | 7.02 ± 4.25 | 2.93 ± 4.43 | 0.02 ± 0.18 | 1.53 ± 2.43 | 0.08 ± 0.89 |
| GCSL | 8.98 ± 5.73 | 7.28 ± 3.43 | 3.12 ± 4.37 | 0.0 ± 0.0 | 0.34 ± 1.82 | 1.18 ± 6.41 |
| GC-TD3BC+RWS (Ours) | 34.19 ± 2.64 | 31.63 ± 3.24 | **49.43 ± 8.19** | **24.39 ± 6.23** | **19.98 ± 5.18** | **48.24 ± 6.24** |
| GC-ReBRAC+RWS (Ours) | 30.95 ± 2.62 | 28.36 ± 3.60 | 28.66 ± 8.12 | 13.42 ± 4.44 | 14.20 ± 4.71 | 30.35 ± 5.99 |
| GC-MCQ+RWS (Ours) | **38.14 ± 2.06** | **31.88 ± 3.01** | 28.46 ± 8.81 | 4.34 ± 3.93 | 2.14 ± 2.63 | 15.18 ± 5.66 |

Table 1: Overall comparisons. The methods with RWS are our proposed sample re-weighting augmented offline RLs. The performance in **bold** indicates the best results according to the t-test with p-value $< 0.05$ with respect to the highest mean.

| Expert-0.5 | FetchPush | FetchPick | HandBlock-Z | HandBlock-XYZ | HandBlock-Parallel | HandEggRotate |
|---|---|---|---|---|---|---|
| TD3BC (RWS) | **34.19 ± 2.64** | **31.63 ± 3.24** | **49.43 ± 8.19** | **24.39 ± 6.23** | **19.98 ± 5.18** | **48.24 ± 6.24** |
| TD3BC | 28.16 ± 3.17 | 28.18 ± 3.40 | 32.81 ± 7.87 | 15.24 ± 5.27 | 14.64 ± 4.64 | 34.00 ± 6.23 |
| ReBRAC (RWS) | **33.04 ± 2.18** | **29.33 ± 3.61** | **35.55 ± 7.82** | 13.42 ± 4.44 | **14.20 ± 4.71** | **30.35 ± 5.99** |
| ReBRAC | 22.77 ± 3.54 | 25.05 ± 3.32 | 28.78 ± 7.30 | 12.50 ± 4.58 | 6.46 ± 2.78 | 19.95 ± 5.21 |
| MCQ (RWS) | 38.14 ± 2.06 | 32.27 ± 2.91 | **28.46 ± 8.81** | 4.34 ± 3.93 | **4.14 ± 2.63** | **15.18 ± 5.66** |
| MCQ | 37.95 ± 1.89 | 31.42 ± 3.03 | 18.80 ± 7.30 | 5.09 ± 3.61 | 0.69 ± 1.28 | 11.44 ± 4.51 |

| Expert-0.3 | FetchPush | FetchPick | HandBlock-Z | HandBlock-XYZ | HandBlock-Parallel | HandEggRotate |
|---|---|---|---|---|---|---|
| TD3BC (RWS) | **30.06 ± 2.86** | **26.19 ± 3.86** | **27.40 ± 7.42** | **10.44 ± 4.30** | 8.81 ± 4.06 | **31.83 ± 6.01** |
| TD3BC | 11.79 ± 3.46 | 16.73 ± 3.36 | 23.20 ± 7.35 | 8.73 ± 3.92 | 7.14 ± 3.05 | 26.73 ± 4.84 |
| ReBRAC (RWS) | **30.83 ± 2.85** | **26.98 ± 3.61** | **25.60 ± 7.29** | **7.54 ± 3.31** | **6.94 ± 3.12** | **13.93 ± 4.43** |
| ReBRAC | 6.90 ± 2.97 | 5.18 ± 1.72 | 21.42 ± 6.98 | 5.19 ± 3.18 | 3.25 ± 2.15 | 9.29 ± 3.11 |
| MCQ (RWS) | 36.12 ± 2.46 | 28.08 ± 3.42 | **25.25 ± 8.18** | **3.81 ± 3.36** | 1.84 ± 2.42 | **14.67 ± 5.42** |
| MCQ | 35.24 ± 2.62 | 30.70 ± 3.24 | 21.89 ± 7.98 | 3.11 ± 2.95 | 1.62 ± 2.34 | 10.54 ± 4.02 |

| Expert-0.1 | FetchPush | FetchPick | HandBlock-Z | HandBlock-XYZ | HandBlock-Parallel | HandEggRotate |
|---|---|---|---|---|---|---|
| TD3BC (RWS) | **28.30 ± 3.27** | **23.77 ± 3.94** | 17.53 ± 6.58 | **3.68 ± 2.52** | **4.71 ± 3.02** | **20.66 ± 5.10** |
| TD3BC | 4.63 ± 2.48 | 3.63 ± 2.12 | 17.88 ± 6.49 | 1.97 ± 1.96 | 1.39 ± 1.67 | 7.83 ± 2.93 |
| ReBRAC (RWS) | **28.25 ± 3.29** | **23.43 ± 3.69** | 15.44 ± 5.99 | 0.61 ± 0.92 | 0.13 ± 0.43 | 2.99 ± 1.70 |
| ReBRAC | 2.02 ± 1.67 | 0.93 ± 0.92 | 14.13 ± 5.79 | 0.70 ± 0.96 | 0.02 ± 0.03 | 2.22 ± 1.40 |
| MCQ (RWS) | **33.14 ± 3.00** | 25.83 ± 3.62 | **24.76 ± 7.82** | 2.23 ± 2.36 | **2.50 ± 1.88** | **8.56 ± 4.43** |
| MCQ | 31.82 ± 3.39 | 26.15 ± 3.56 | 20.91 ± 7.75 | 3.22 ± 2.66 | 0.22 ± 0.27 | 4.50 ± 2.69 |

Table 2: RWS with different dataset qualities. Performance metrics (mean ± standard error) were calculated and analyzed using a T-test to determine the best performer for each task and dataset quality. The most effective configurations, as determined by these T-tests, are highlighted in bold. **(The numbers 0.5, 0.3, and 0.1 denote the ratios of expert demonstration data in the offline dataset.)**

porating RWS into offline goal-conditioned RL substantially enhances policy performance across multiple tasks.

## 5.2 Dataset Quality Impacts

We evaluate how dataset quality influences offline goal-conditioned RL algorithms when combined with Reachability Weighted Sampling (RWS). Specifically, we create three datasets by varying the ratio of expert demonstrations (0.5, 0.3, and 0.1) and train each algorithm with 10 random seeds. Table 2 reports the mean

performance, along with standard errors and T-test comparisons for each task configuration. Overall, we observe that RWS consistently boosts the performance of action-regularized methods (TD3BC and ReBRAC) across different dataset qualities. For instance, TD3BC with RWS achieves substantial gains in HandBlock-XYZ and HandEggRotate at an expert ratio of 0.5. In contrast, its effectiveness diminishes when the ratio is reduced to 0.1. Moreover, RWS yields notable improvements for MCQ, particularly in high-dimensional tasks and with higher fractions of expert trajectories. These findings underscore a positive correlation between expert ratio and performance, suggesting that RWS has diminishing returns when the dataset contains relatively few expert demonstrations. The results in Table 2 indicate that RWS can enhance policy learning in offline goal-conditioned RL by prioritizing potentially reachable goals, but the magnitude of improvement depends on the proportion of expert data in the offline dataset.

## 5.3    Comparison of resampling methods

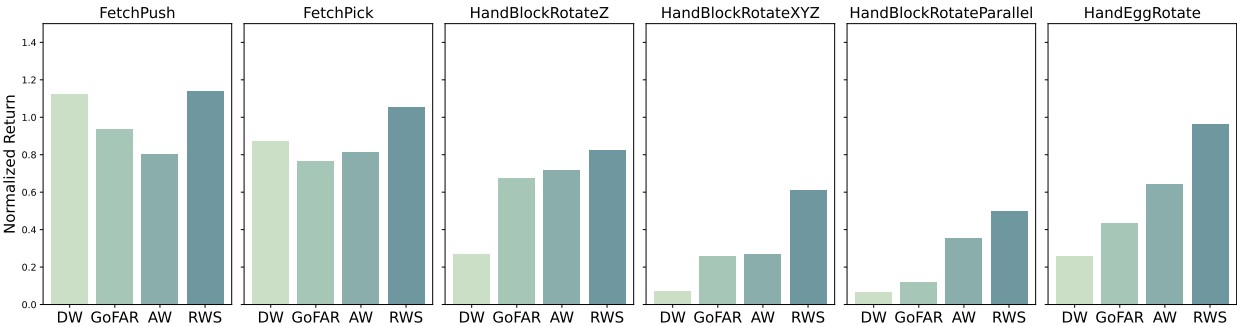

Figure 6: Weighted sampling methods comparisons : DW, GoFAR, AW, and our RWS. We use TD3BC as the base algorithm. We report the performance as average normalized returns: the accumulated return divid by the best performance in the dataset. for each resampling technique. We trained each method with 10 random seeds.

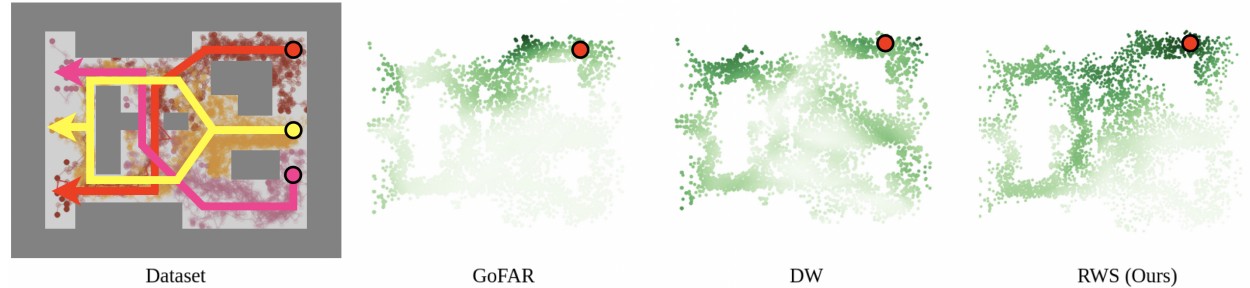

Figure 7: The visualization of the learned goal-conditioned sampling weight. The left most plot visualizes the 2D maze datset and 3 types of goal-conditioned trajectories. We define the navigation agent's starting position on the top-right corner (red dot with black circle) and use DW, GoFAR and RWS to estimate what areas are reachable. The following plots are the visualization of the weights of different methods (the darker color indicates higher reachability).

**Which weighted sampling for offline GCRL?**    To further validate our approach, we compare Reachability-Weighted Sampling (RWS) with other weighted sampling techniques for offline GCRL. In this experiment, we use GC-TD3BC as the base algorithm and integrate it with four resampling methods: Density-Weighting (DW), Advantage-Weighting (AW), GoFAR, and our proposed RWS. Each method is trained with 10 random seeds. Performance is measured as the average normalized return, where a larger

value indicates a better policy. Figure 6 shows that RWS consistently achieves the highest normalized return across all tasks. This improvement is particularly pronounced in tasks such as HandBlockRotateXYZ, HandBlockRotateParallel, and HandEggRotate. These results highlight the robustness of RWS in handling different levels of complexity in robotic manipulation tasks.

**Sample reweighting comparisons**   To further analyze how different weighting techniques assign weights to goals, we created a 2D maze dataset. In this environment, each point's location is defined by its x-y coordinates, which serve as the observation, while the target's x-y coordinates define the goal. The leftmost figure in Figure 7 shows the offline dataset, composed of three trajectory types (from top-right to bottom-left, middle-left to middle-right, and bottom-left to top-right). Given a common starting position $s$ at the top-right corner, we sample all positions as potential goals to form state-goal pairs and visualize the reachability sampling weights produced by different methods. As shown in Figure 7, GoFAR tends to assign higher weights to positions close to the starting point. DW, however, exhibits unpredictable behavior: it assigns high weights both to the endpoints of trajectories (allowing the agent to reach three different maze endpoints) and to clearly unreachable positions (such as the starting positions on the right), while assigning low weights to reachable areas in the middle. In contrast, our RWS method generates the most meaningful reachability weights by assigning lower weights to unreachable regions (e.g., the bottom-right area) and higher weights to regions where goals are reachable. This balanced weighting promotes effective goal stitching in the maze.

Overall, the results in Figures 6 and 7 strongly indicate that RWS offers a superior resampling strategy for offline RL. It not only enhances the performance of standard algorithms like TD3BC but also provides more consistent and interpretable weighting across the state space.

## 6   Conculusion

This paper proposes Reachability Weighted Sampling (RWS), a plug-and-play strategy for offline goal-conditioned reinforcement learning (GCRL). RWS leverages a positive–unlabeled (PU) learning-based classifier, which captures goal reachability by mapping goal-conditioned Q-values to classification scores. These scores guide a sampling priority that effectively focuses on transitions that facilitate goal achievement. We integrate RWS with standard offline RL methods (Levine et al., 2020; Agarwal et al., 2023) and evaluate it on six simulated robotic manipulation tasks, including Fetch and dexterous hand environments, achieving new state-of-the-art results. Notably, RWS yields nearly a 50% improvement in HandBlock-Z compared to a strong baseline, demonstrating its potential for data-efficient offline GCRL. Despite these promising findings, RWS depends on reliable Q-value estimation to accurately classify reachability. Future work should take a deeper study of RWS and contrastive GCRLs to unify the weighted goal sampling framework with the contrastive GCRL frameworks (Eysenbach et al., 2022; Zheng et al., 2023; Eysenbach et al., 2020), and explore its applicability in real-world scenarios where data coverage is limited.

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
