# OpenReview forum: "Reachability Weighted Offline Goal-conditioned Resampling"
_TMLR — Rejected by TMLR_

### Review · Reviewer_ksVG · 2025-05-12

**Summary Of Contributions:**

This paper introduces a plug and play solution for offline goal-conditioned reinforcement learning. The method introduces
weighted sampling that prioritizes state-goal-action pairs with achievable goals. This allows this method to be more effective in comparison to randomly sampling state-goal-action pairs. Reachability scores are based a PU classifier. The method appears to be able to be integrated with most common RL algorithms.

**Audience:**

Yes

**Claims And Evidence:**

Yes

**Requested Changes:**

Please kindly address the weaknesses.

**Strengths And Weaknesses:**

Strengths:
The paper is well written and the results show that RWS outperforms other standard methods.
The approach is conceptually simple and seems to be able to be used in a plug and play fashion seamlessly.

Weaknesses:
The plots in Figure 6 do no display the mean and standard deviation of the results. This would help us quantify
the variance in the methods.

There is no discussion regarding the computational complexity of the method, especially in comparison to the baselines, or the potential of training instabilities. For instance a poorly learned q-value function can lead to poor classification and instability during this two step training.

What is the effect of the choice of the prior probability $\eta_p$? I believe some justification is needed for the choice of its value. There should be some discussion of how such parameters affect the results.

Why not also compare with HER, since it can be applied in an offline setting?

How did the value network get initialized in Algorithm 1?

Just to clarify, I assume that the t-test used in Table 2 is performed between two treatments one with the RWS and one without. How is the t-test performed in Table 1? Do you account for multiple comparisons?

There are some typos in the paper that need to be fixed.

---

### Review · Reviewer_ybZi · 2025-05-18

**Summary Of Contributions:**

This paper proposes a weighted sampling method in offline goal-conditioned RL with "reachability score" as the priority. The authors first observed that the random sampling of goals can cause over-conservatism and deteriorate the performance. They then construct a positive-unlabeled dataset by hindsight experience replay and random sampling and apply non-negative PU learning on that. The experiments on fetching tasks and hand manipulation tasks show that the proposed method can be plugged into existing offline GCRL methods and enhance their performances.

**Audience:**

Yes

**Claims And Evidence:**

Yes

**Requested Changes:**

- Answer the above questions: explain the motivations of reachability score parameterization and experiment settings more clearly.
- Add ablations for hyper-parameters.

**Strengths And Weaknesses:**

**strengths**

- Overall, the paper is very clearly written and easy to follow.
- The idea of using a scalar score to weigh the sampling probability is straightforward, which also proves to be effective.
- The experiments are extensive in terms of the tasks and baselines. The results show that the proposed method can be plugged into existing algorithms and boost their performances.

**weaknesses**

I don't find major weaknesses for this paper. Here are some of my questions:

- Why is the reachability score computed based on the Q value instead of the state-action pair? In my understanding, the score $C(s,a,g)$ should be more intuitive and expressive than $C(Q(s,a,g))$.
- What are the proportions of posititve and unlabeled data in experiments?
- How do you select the $\eta$ of positive and unlabeled data in PU learning?

minor typos:
- Last line of page 3, incorrect citation format.
- In eq.(6), the order of $s,a,g$ in Q function is inconsistent: $Q(s,g,a)$ and $Q(s,a,g')$

---

### Review · Reviewer_rXZT · 2025-06-22

**Summary Of Contributions:**

This paper introduces Reachability Weighted Sampling (RWS), a novel method to improve offline goal-conditioned reinforcement learning (GCRL) by prioritizing transitions that are more likely to lead to successful goal achievement. Existing approaches rely on uniformly sampling state-goal pairs, which often results in training on unreachable or suboptimal transitions, degrading policy quality. RWS addresses this by training a reachability classifier using positive-unlabeled (PU) learning, where reachable transitions are generated via hindsight relabeling and unlabeled transitions are sampled randomly. The classifier uses goal-conditioned Q-values to estimate reachability, and these estimates are then transformed into sampling weights, ensuring that learning focuses on more promising experiences.

**Audience:**

Yes

**Broader Impact Concerns:**

No concern.

**Claims And Evidence:**

Yes

**Requested Changes:**

Please see weaknesses in the above section. In summary, below are some questions that can help strengthen the paper in my opinion:

1. While I believe that the motivation around solving manipulation using the proposed goal sampling component is well supported by the results, the authors do claim that this can be generally useful for other tasks as well. Can the authors provide some specific discussion around what other tasks outside of the manipulation domain might benefit from RWS?
2. In low-data or noisy environments (such as with real-world robots), the Q-function may be inaccurate, leading to poor reachability estimates and suboptimal sampling. How will this affect the accuracy of RWS?

**Strengths And Weaknesses:**

**Strengths:**

1. I find the paper well-written and easy to follow. The authors made it very clear in the introduction what the scope and key contributions of this paper are.
2. Leveraging positive-unlabeled (PU) learning for estimating reachability from goal-conditioned Q-values is novel and theoretically grounded.
3. RWS is designed to be modular and easy to integrate with existing offline RL algorithms without needing architectural changes.
4. The paper provides extensive experimental validation across six diverse robotic tasks, including high-dimensional dexterous hand manipulation.

**Weaknesses:**

1. The effectiveness of RWS seems to depend heavily on the quality of the goal-conditioned Q-function, which is used as the basis for reachability classification. When the Q-function is be inaccurate, this may lead to poor reachability estimates and suboptimal sampling, which might affect the effectiveness of RWS.
2. While the simulated robotics tasks are well-established benchmarks, the paper lacks validation in real-world settings, where offline data may be much noisier or less structured.
3. The generalization of RWS to real-world domains (e.g., vision-based or natural language goals) remains untested.

---

### Decision · Action_Editor_6MxD · 2025-08-14

**Recommendation:** Reject

**Additional Comments:**

The authors are encouraged to address the questions from all reviewers in a revision, and resubmit.

**Audience:**

Yes

**Audience Explanation:**

The reviewers generally agree that TMLR's audience would be interested in knowing the findings of this paper.

**Claims And Evidence:**

No

**Claims Explanation:**

The reviewers held the opinion that the claims are not yet sufficiently supported by convincing and clear evidence: Reviewer **rXZT** had concerns that the proposed RWS approach appears to depend on the quality of the goal-conditioned Q-function, making it challenging to apply RWS in low-data or noisy environments. They also had questions on the generalizability of the approach to real-world tasks, which requires additional empirical validations. Reviewer **ksVG** had a series of questions regarding the detailed design choices made in the empirical evaluations (e.g., standard deviation of the results, effect of prior probability, initialization of the value network), they also requested discussion on the computational complexity of the method, and additional comparison with HER. Reviewer **ybZi** found no major weaknesses, but asked a few clarification questions.

The authors did not provide a revision or a rebuttal to clarify the questions, or to provide the requested experimental validations.

**Resubmission Of Major Revision:**

The authors may consider submitting a major revision at a later time.